Soil microbial diversity and functional capacity associated with the production of edible mushroom Stropharia rugosoannulata in croplands

Tang Shaojun 1
Fan Tingting 2
Jin Lei 1
Lei Pin 1
Shao Chenxia 1
Wu Shenlian 1
Yang Yi 1
He Yuelin 1
Ren Rui 1
Xu Jun hnswsw_xj@163.com 1
1 Hunan Institute of Microbiology , Changsha , china
2 College of Forestry, Central South University of Forestry & Technology , Changsha , China
Cao Yunpeng
Electronic publication date: 2022 Oct 3
Publication date: 2022
Volume: 10
Electronic Location ID: e14130
Received 2022 Aug 15; Accepted 2022 Sep 6
Copyright: ©2022 Tang et al.
Copyright year: 2022
Copyright holder: Tang et al.
License: This is an open access article distributed under the terms of the Creative Commons Attribution License, which permits unrestricted use, distribution, reproduction and adaptation in any medium and for any purpose provided that it is properly attributed. For attribution, the original author(s), title, publication source (PeerJ) and either DOI or URL of the article must be cited.
License URL: https://creativecommons.org/licenses/by/4.0/

Keywords: Soil physicochemical properties, Stropharia rugosoannulata, Soil bacterial communities, Soil fungal communities, Functional analysis

Funding: National Natural Science Foundation of China 2022JJ40233 Changsha Key R&D Projects kh2201216 Hunan Province Key R&D Projects 2019NK2192 Hunan Provincial Natural Science Foundation of China 2020JJ4049 This research was funded by the National Natural Science Foundation of China (2022JJ40233), the Changsha Key R&D Projects (kh2201216), the Hunan Province Key R&D Projects (2019NK2192) and the Hunan Provincial Natural Science Foundation of China (2020JJ4049). These funders guided our experiments.

==============================
In recent years, a rare edible mushroom Stropharia rugosoannulata has become popular. S. rugosoannulata has the characteristics of easy cultivation, low cost, high output value, and low labor requirement, making its economic benefits significantly superior to those of other planting industries. Accumulating research demonstrates that cultivating edible fungus is advantageous for farming soil. The present experiment used idle croplands in winter for S. rugosoannulata cultivation. We explored the effects of S. rugosoannulata cultivation on soil properties and soil microbial community structure in paddy and dry fields, respectively. We cultivated S. rugosoannulata in the fields after planting chili and rice, respectively. The results showed that Chili-S. rugosoannulata and Rice-S. rugosoannulata planting patterns increased the yield, quality and amino acid content of S. rugosoannulata. By analyzing the soil properties, we found that the Chili-S. rugosoannulata and Rice-S. rugosoannulata cropping patterns increased the total nitrogen, available phosphorus, soil organic carbon, and available potassium content of the soil. We used 16s amplicons for bacteria and internal transcribed spacer (ITS) region for fungi to analyze the microbial communities in rhizosphere soils. Notably, S. rugosoannulata cultivation significantly increased the abundance of beneficial microorganisms such as Chloroflexi, Cladosporium and Mortierella and reduce the abundance of Botryotrichumin and Archaeorhizomyces. We consider S. rugosoannulata cultivation in cropland can improve soil properties, regulate the community structure of soil microorganisms, increase the expression abundance of beneficial organisms and ultimately improve the S. rugosoannulata yield and lay a good foundation for a new round of crops after this edible mushroom cultivation.

Introduction

The edible fungi industry is an important industry in the global planting industry, and most edible fungi extracts have biological activities such as antioxidants (Pejin et al., 2019; Tesanovic et al., 2017). S. rugosoannulata is a rare edible mushroom originally discovered by Professor Murrill in the United States (Yan et al., 2020). Because of its high economic and ecological value, S. rugosoannulata was introduced to China as early as the 1980s and has been widely planted (Yan et al., 2020). Experimental pharmacological studies have shown that S. rugosoannulata has various biological activities such as antitumor, antibacterial, antioxidant, and immunomodulatory (Wu et al., 2011; Wu et al., 2012). In recent years, the demand for S. rugosoannulata has been increasing due to its delicious and pharmacological properties, so it is urgent to increase its production  (Liu et al., 2020).

Crop rotation appears to be an effective method for boosting S. rugosoannulata output (Peltonen-Sainio, Jauhiainen & Lehtonen, 2016). This is due to long-term inappropriate use of the land, such as diffuse irrigation, excessive use of fertilizers, pesticides, etc., can lead to damage to the soil structure, imbalance of nutrient ratios, degradation, and acidification  (Gong et al., 2018). Rotation can increase the soil organic carbon and total nitrogen content, improve soil fertility, and boost crop output (Kuht et al., 2019; Eremeev et al., 2020). The mycelium produced by growing edible mushrooms contains various trace elements (Koutrotsios et al., 2020). The residual mycelium is rich in crude protein, crude fat, calcium, phosphorus, and other organic substances (Yun et al., 2021). Some bioactive substances secreted by the mycelium in the production process can decompose complex organic matter, inhibit some soil-borne diseases and promote plant growth (Grimm & Wösten, 2018). The texture of the mushroom residue is loose and has a solid ability to absorb water, which can be further decomposed into humus with good air and water storage capacity and can effectively improve the soil (Lipiec et al., 2021). This indicates that mushroom residue can well regulate the soil properties of agricultural land and enhance soil fertility. In addition, crop cultivation in cropland also contributes to mushroom culture (De la Fuente, Beyer & Rinker, 1998). That is because the rotation of different crops on the same soil can significantly improve the physical and chemical properties of the soil, regulate soil fertility, and ultimately achieve increased production and income (Kumar et al., 2020).

Chili and rice are two important economic crops in China’s Hunan Province, and both have a positive impact on soil quality. Therefore, it seems to be a good choice to cultivate S. rugosoannulata in winter on unused acreage. Studies have indicated that using chili as a crop rotation can greatly enhance soil nitrogen content and minerals after planting, as well as increase carbon fixation in soil (Hao et al., 2011). Hahm et al. (2017) discovered that using chili as a crop rotation boosted the expression abundance of Chloroflexi in soil while decreasing salt accumulation. In addition to regulating soil microorganisms, chili inhibits the development and proliferation of Meloidogyne spp., according to research conducted by Bafokuzara (1983). Some experiments have found that the implementation of paddy upland rotations increases the stability of soil physical properties, improves the bulk density of soil, reduces damage to soil structure, and increases soil water stable aggregate (Hou et al., 2018). On the other hand, unlike dryland soils alone, soils in paddy upland rotations change the chemical form of the nutrients in them due to alternating wet and dry, anaerobic and aerobic conditions (Zhou et al., 2014). Crop rotation can effectively improve soil utilization, improve soil structure through the roots of different crops, reduce soil consolidation and degradation, and increase soil nutrient content (Hamza & Anderson, 2005).

Currently, the majority of research use horticultural crops in rotation with chili or rice. Few studies have investigated the influence of edible fungus as rotation crops on soil characteristics. In this experiment, S. rugosoannulata was used for the first time to rotate with chili and rice. Additionally, S. rugosoannulata was cultivated by maximizing the use of idle farmland during the winter, and it was investigated whether the cultivation process as a whole could regulate soil properties and soil microorganisms. This study will contribute to the commercial production of S. rugosoannulata and enhance the utilization of vacant agricultural land in winter.

Materials & Methods

Experimental design

Three experimental plots of the same size (two hectare) in the experimental field of the Hunan Institute of Microbiology (longitude: 112.98, Latitude: 28.12) were selected. These three plots have the same size and adjacent positions. The experiment began in April 2020. The chili variety used in the experiment was (Capsicum annuum L.), ‘SJ11-3’. The rice variety used in the experiment was “Zhongjiazao 17”. Cultivation of S. rugosoannulata in vacant fields (NC), cultivated S. rugosoannulata after harvested chili (Chili-S. rugosoannulata) and rice (Rice-S. rugosoannulata) crops. In April 2020, chili was planted in the experimental field of the Chili-S. rugosoannulata group. Chili was harvested in October 2020. In June, rice was planted in the experimental field of the Rice-S. rugosoannulata group. Rice was harvested in September 2020. In mid-November 2020, the ground of all test fields was leveled and all crop residues were removed and not plowed into the soil. The quicklime was used to disinfect and kill insects on the fields. Straw, corncob, and chaff was soaked in 3% lime water for two days, then mixed in an 8:1:1 ratio and distributed equally with a thickness of 30 cm on all experimental fields. It was then paved with the S. rugosoannulata spawn (300 kg/mu) and covered with approximately 2–3 cm of soil. After covering the soil, it was covered with a layer of rice straw (the covering layer is not visible, the thickness is 3–4 cm). Three to four months were required for the development of the fungi to fruiting bodies. The root soil from the three experimental fields was sampled the same year in mid–November, next year in January and March to see the effect of fungal growth on the soil quality. We used the five-point sampling method to produce a composite sample by combining five independent soil cores from around and within the croplands (Liu et al., 2021a). Six composite samples were taken from each experimental field. Fresh samples were sealed in plastic bags and stored in a −80° freezer.

Determination of amino acid content in S. rugosoannulata

After crushing S. rugosoannulata through a 50-mesh sieve (the aperture is 0.3 mm), add 2 ml of concentrated hydrochloric acid, mix 2 to 3 drops of phenol and evacuate heat melt the glass hydrolysis tube while keeping the hydrolysis tube in a high vacuum state, and complete the sealing. Place the hydrolysis tube in an oven at 110 °C. After hydrolyzing for 24 h, take it out and cool it to room temperature. Transfer the hydrolyzed sample to a 25ml volumetric flask. Wash the hydrolysis tube repeatedly with pure water and fill the volume up to the mark. After the constant volume of the sample is passed through a 0.22 µm microporous membrane, put it into the sample vial to be tested. Analysis and identification of amino acids in biological tissues utilizing a high-performance liquid chromatograph (Model 1100; Agilent, USA). The chromatographic conditions are: chromatographic column: Zorbax Eclipse AAA C18 column (75 mm × 4.6 mm, 3.5 µm, Agilent, USA); mobile phase: A is 40 mmol/L NaH2PO4 solution, pH 7.8; B is acetonitrile-methanol-water (volume ratio of 4.5:4.5:1). Mobile phase gradient elution program: 0.0–1.0 min (B: 0%), 1.0–9.8 min (B: 0%–57%), 9.8–10.0 min (B: 57% ∼100%), 10.0–12.0 min (B: 100%), 12.0–12.5 min (B: 100% ∼0%), 12.5–14.0 min (B: 0%); flow rate: two mL/min; column temperature: 40 °C; injection volume: 18 µl.

Determination of soil total nitrogen

A total of 1.00 g air-dried soil sample with 100-mesh sieve (the aperture is 0.15 mm) was weighed and carefully placed in the Kjeldahl flask. A small amount of distilled water was added for wetting and washing. Then, 2 g of the prepared mixed catalyst was added, and 5 ml of concentrated sulfuric acid was added to cover the bottle cap with a small funnel. Place the prepared bottle on the digester at 375 °C for 1 h. The end of cooking is a clear light blue mixture of the cooking furnace. After digestion, the Kai-type bottle was removed, standing for cooling, shaking in the middle for several times to prevent precipitation, and the small funnel was taken down. The small funnel was washed with distilled water to make all the solution in the small funnel enter the Kai-type bottle. The supernatant was taken and diluted to adjust the pH value. The total nitrogen content in the solution was measured by a flow analyzer (AA3Auto Analyzer 3 Continuous-Flow Analysis-3; SEAL, Germany), and then converted into soil total nitrogen content (Dane, Topp & Campbell, 2002).

Determination of soil available phosphorus

Weigh 2.5 g air-dried soil sample which has passed through a 20-mesh sieve (the aperture is 0.85 mm), add 50 ml NaHCO3 (0.5 mol L−1) solution and phosphorus-free activated carbon, shake for 30 min, and filter. Remove the filtrate and add 2-6 dinitrophenol color reagent, adjust the solution to slightly yellow with diluted acid, add Mo-Sb antimony reagent and read the absorbance value at 720 nm wavelength (Bray & Kurtz, 1945).

Determination of soil organic carbon

Weigh a soil sample that has been quantitatively passed through an 80-mesh sieve (the aperture is 0.18 mm) placed in a dry hard test tube, with the transfer tube to accurately add 0.8000 mol/L (1/6K2CrO7) standard solution 5 mL, accurately add concentrated sulfuric acid 5 ml fully shake. The test tube was placed in a wire cage, and then the wire cage was placed in an oil bath to heat. After being placed, the temperature should be controlled at 170–180 °C. When the liquid in the test tube boils and bubbles occur, it starts to count, boils for 5 min, removes the test tube, slightly cools, and wipes out the external oil of the test tube. Take out the cooling. After cooling, all the contents in the test tube were carefully transferred into 250 ml triangle bottle, so that the volume of the contents in the triangle bottle was 60–70 ml. Keep sulfuric acid concentration of 1∼1.5 mol/l, the solution color should be orange or light yellow. Then add o-norphine indicator 3∼4 drops, titrate with 0.2 mol/l standard ferrous sulfate (FeSO4) solution, the solution from yellow through green, light green to brown is the end point (Ciavatta et al., 1991).

Determination of soil moisture

Weigh a 5 g soil sample, put it in an oven at 105 °C, and bake it until the weighing remains constant before and after twice. Loss of moisture after drying is the soil moisture content.

Determination of soil available potassium

Weigh a 5 g air-dried soil sample which has passed through a 20-mesh sieve (the aperture is 0.85 mm), add 50 ml NH4OAc (1 mol L−1) solution, shake for 30 min, filter, and take the filtrate for measurement on a flame photometer (Jackson, 2005).

Soil DNA extraction and library preparation

Use EZNA® soil DNA kit (Omega Bio-tek, Norcross, GA, USA) to extract DNA from soil samples. The primer pair 338F (5′-ACTCCTACGGGAGGCAGCA-3′) and 806R (5′-GGACTACHVGGGTWTCTAAT-3′) was used to amplify the bacterial 16S V3+V4 region of extract DNA. The fungal ITS2 region of extract DNA was amplified using primer pair ITS2F (5′-GCATCGATGAAGAACGCAGC-3′) and ITS2R (5′-TCCTCCGCTTATTGATATGC-3′). PCR reaction conditions were 25 cycles of 95 °C 5min, 95 °C 30 s, 50 °C 30 s, 72 °C 40 s, and finally 72 °C for 7 min, and stored at 4 °C. The PCR products obtained were quantified by electrophoresis (ImageJ software) and mixed in a mass ratio of 1:1. Column purification was performed using OMEGA DNA purification columns (bio-tek, Doraville, GA, USA). Finally, use 1.8% agarose gel, electrophoresis at 120 V for 40min, and cut the gel to recover the target fragment.

Illumina sequencing and microbial diversity analysis

The samples were sequenced on the Illumina MiSeq platform, the original data were spliced (FLASH, version 1.2.11) (Magoč & Salzberg, 2011), using Trimmomatic (version 0.33) to filter the quality of the raw data (Bolger, Lohse & Usadel, 2014), then use Cutadapt (version 1.9.1) to identify and remove primer sequences, and then use USEARCH (version 10) to splice paired-end reads and remove chimeras (UCHIME, version 8.1) finally obtained high-quality sequences for subsequent analysis (Edgar et al., 2011). Sequences with a similarity of more than 97% were clustered as an OTU (USEARCH, version 10.0), and 0.005% of all sequences sequenced were used as a threshold to filter OTUs (Edgar, 2013). Bacteria were annotated using the Silva database (Release128, http://www.arb-silva.de), and fungi were annotated using Unite (Release 7.2, http://unite.ut.ee/index.php). Species annotation was performed using RDP Classifier with a confidence threshold of 0.8 (version 2.2, https://sourceforge.net/projects/rdp-classifier/). Mothur version 1.30 (http://www.mothur.org/) for Alpha diversity index analysis. All sample sequences were uploaded to the NCBI database, with sample numbers ranging from SRX13373992 to SRX13374011.

Correlation and predictive analysis of microbial function

Correlation analysis and mapping of soil quality and soil microorganisms using the R language vegan (v2.3) package. RDA (Redundancy analysis) is a sorting method developed based on correspondence analysis; RDA analysis is based on a linear model, mainly used to reflect the relationship between flora or samples and environmental factors. The size of the first axis of Lengths of a gradient in the analysis result should be less than 4.0. PICRUSt2 (Phylogenetic Investigation of Communities by Reconstruction of Unobserved States 2) software was used to predict the abundance of marker functional gene sequences in samples. PICRUSt2 is a computational method that utilizes marker gene data and a reference genome database to predict the functional composition of environmental microbes, based on IMG microbial genome data to predict microbial communities’ functional potential during phylogeny through phylogeny phylogenetic-functional correlations. Based on the software’s reference genome data, both 16S rRNA sequences and ITS sequences can be used for functional prediction (Langille et al., 2013).

Determination of S. rugosoannulata quality

Take the fruiting body of S. rugosoannulata, measure the diameter and length of its stipe, and determine whether it is hollow. 5 kg of fruiting bodies were randomly selected in each experimental field to evaluate the quality of S. rugosoannulata.

Data analysis

All data in this study were analyzed using SPSS 22.0, expressed as mean  ± standard deviation of the mean (SD). The difference between the means of the groups was analyzed using a one-way ANOVA pair and evaluated using Tukey multiple comparisons. p < 0.05 was regarded as a significant difference.

Results

Quality and yield of S. rugosoannulata in different croplands

First, we determined the quality of S. rugosoannulata. As shown in Fig. 1, the mushroom stipe of S. rugosoannulata grown in the NC group was elongated and hollow (Fig. 1A). However, compared with the NC group, the stipe of S. rugosoannulata grown in rotation with chili and rice was significantly thicker (Figs. 1B and 1C). Among them, S. rugosoannulata which was rotated with rice, has the thickest mushroom stipe, the largest volume, and the best quality. The yield of S. rugosoannulata in different croplands was shown in Fig. 1D. Compared with NC, the yield of Chili-S. rugosoannulata and Rice-S. rugosoannulata was significantly higher (p < 0.05). Among them, S. rugosoannulata, which was rotated with rice, has the highest yield. It reached about 240 kg/hm2.

Figure 1 The quality and yield of S. rugosoannulata in different croplands.

(A) S. rugosoannulata produced in the NC group; (B) S. rugosoannulata produced in the Chili-S. rugosoannulata group; (C) S. rugosoannulata produced in the Rice-S. rugosoannulata group; (D) Yield per mu of S. rugosoannulata in different croplands. Data are mean ± SD (n = 3) and analyzed by one-way ANOVA. *p < 0.05.

Amino acid content of S. rugosoannulata in different croplands

Table 1 shows the analysis of amino acid content in S. rugosoannulata. For the S. rugosoannulata in Chili-S. rugosoannulata, there was a substantial increase in the level of Thr, Ser, Glu, Gly, Ala, Val, Ile, Leu, Tyr, His, Lys, and Pro (p < 0.05) relative to the control S. rugosoannulata. Meanwhile, for the S. rugosoannulata in Rice-S. rugosoannulata, the levels of 16 types of amino acids increased substantially (p < 0.05) in comparison to the control S. rugosoannulata.

Table 1 The amino acid content of S. rugosoannulata in different croplands.

Amino acid (µg/mg)	NC	Chili-S. rugosoannulata	Rice-S. rugosoannulata	
Asp	1.670 ± 0.039b	1.589 ± 0.053b	2.060 ± 0.041a	
Thr	0.901 ± 0.023c	0.968 ± 0.017b	1.128 ± 0.050a	
Ser	0.892 ± 0.025c	0.984 ± 0.024b	1.091 ± 0.038a	
Glu	2.362 ± 0.222b	3.231 ± 0.036a	3.067 ± 0.085a	
Gly	0.715 ± 0.018c	0.813 ± 0.029b	1.071 ± 0.021a	
Ala	0.975 ± 0.008c	1.367 ± 0.102b	1.502 ± 0.030a	
Val	0.972 ± 0.021c	1.104 ± 0.034b	1.252 ± 0.022a	
Met	0.313 ± 0.019b	0.308 ± 0.020b	0.480 ± 0.042a	
Ile	0.667 ± 0.021c	0.801 ± 0.015b	0.905 ± 0.022a	
Leu	1.225 ± 0.038b	1.487 ± 0.196a	1.557 ± 0.053a	
Tyr	0.538 ± 0.022b	0.620 ± 0.020a	0.594 ± 0.015a	
Phe	0.855 ± 0.009b	0.940 ± 0.023b	1.043 ± 0.082a	
His	2.967 ± 0.085c	2.235 ± 0.035b	5.881 ± 0.159a	
Lys	0.960 ± 0.025c	1.078 ± 0.031b	1.388 ± 0.026a	
Arg	0.746 ± 0.015b	0.783 ± 0.016b	0.921 ± 0.036a	
Pro	0.486 ± 0.015b	0.558 ± 0.037a	0.599 ± 0.015a	
Notes.

a,b,c Mean values with different letters are significantly different from each other (p < 0.05).

Soil properties of different croplands

We found that compared with the NC control group, the Chili-S. rugosoannulata and Rice-S. rugosoannulata rotations significantly improved the total nitrogen (Fig. 2A), available phosphorus (Fig. 2B), soil organic carbon (Fig. 2C) and available potassium (Fig. 2D) content in the soil before, during and after cultivating S. rugosoannulata (p < 0.05). Among them, the soil of S. rugosoannulata and rice rotation has the highest content of total nitrogen, available phosphorus, soil organic carbon, and available potassium. Compared with the NC control group, the Chili-S. rugosoannulata and Rice-S. rugosoannulata rotations significantly improved the soil moisture before and after cultivating S. rugosoannulata and it is the highest in group Rice-S. rugosoannulata (p < 0.05). However, during the cultivating of S. rugosoannulata, there was no significant difference in soil moisture among the three groups (Fig. 2E) (p > 0.05).

Figure 2 Soil properties in different croplands.

(A) The total nitrogen content of Soil before, during, and after cultivating S. rugosoannulata under different rotation patterns; (B) The available phosphorus content of Soil before, during, and after cultivating S. rugosoannulata under different rotation patterns; (C) The organic carbon content of Soil before, during, and after cultivating S. rugosoannulata under different rotation patterns; (D) The available potassium content of Soil before, during, and after cultivating S. rugosoannulata under different rotation patterns; (E) The soil moisture before, during, and after cultivating S. rugosoannulata under different rotation patterns. Data are mean ± SD (n = 3) and analyzed by one-way ANOVA. *p < 0.05.

The soil rhizosphere microbial community diversity in different croplands

After cultivating S. rugosoannulata, Chili-S. rugosoannulata, and Rice-S. rugosoannulata, two soil rhizosphere microorganisms ACE index, Chao index, Shannon index, and Simpson index were significantly higher than the NC group (p < 0.05) (Figs. 3A–3D).

Figure 3 α-diversity of soil rhizosphere microorganisms after cultivating S. rugosoannulata in different croplands.

(A) ACE index; (B) Chao index; (C) Shannon index; (D) Simpson index of soil rhizosphere microorganisms after cultivating S. rugosoannulata.

The structure of soil rhizosphere microbial community in different croplands

At the phylum level, after cultivating S. rugosoannulata, Proteobacteria, Acidobacteria, Actinobacteria, and Chloroflexi were the main microorganisms in the three groups, accounting for more than 70% of the total abundance (Fig. 4A). Compared with the NC group, the abundance of Proteobacteria, Bacteroidetes, and Actinobacteriain in the Chili-S. rugosoannulata group after cultivating S. rugosoannulata significantly decreased, while the abundance of Chloroflexi flora increased significantly (p < 0.05). Compared with the NC group, the abundance of Bacteroidetes and Actinobacteriain in the Rice-S. rugosoannulata group after cultivating S. rugosoannulata significantly decreased, while the abundance of Chloroflexi increased significantly (p < 0.05) (Figs. 4B–4E).

Figure 4 Rhizosphere bacterial community structure at phylum level after cultivating S. rugosoannulata in different croplands.

(A) Relative abundance of rhizosphere microbiota at the phylum level after cultivating S. rugosoannulata; (B) percentage of Proteobacteria in each sample from the three groups after cultivating S. rugosoannulata; (C) percentage of Bacteroidetes in each sample from the three groups after cultivating S. rugosoannulata; (D) percentage of Chloroflexi in each sample from the three groups after cultivating S. rugosoannulata; (E) percentage of Actinobacteria in each sample from the three groups after cultivating S. rugosoannulata; (F) Relative abundance of rhizosphere microbiota at the genus level after cultivating S. rugosoannulata; (G) percentage of Sphingomonas in each sample from the three groups after cultivating S. rugosoannulata; (H) percentage of Candidatus_Solibacter in each sample from the three groups after cultivating S. rugosoannulata; (I) percentage of Granulicella in each sample from the three groups after cultivating S. rugosoannulata; (J) percentage of Gemmatimonas each sample from the three groups after cultivating S. rugosoannulata. Data are mean ± SD (n = 6) and analyzed by one-way ANOVA. *p < 0.05.

At the genus level, the bacterial community composition in the rhizosphere after cultivating S. rugosoannulata is shown in Fig. 4F. After cultivating S. rugosoannulata, compared with the NC group, the abundance of Sphingomonas and Burkholderia-Caballeronia-Paraburkholderia and Granulicella in the Chili-S. rugosoannulata group was significantly reduced, while the abundance of Candidatus Solibacter was significantly increased (p < 0.05). Rice-S. rugosoannulata group significantly increased the abundance of Sphingomonas, C. Solibacter and Gemmatimonas (p < 0.05) (Figs. 4G–4J).

The structure of soil rhizosphere fungal community in different croplands

At the phylum level, after cultivating S. rugosoannulata, Basidiomycota, Ascomycota, and Mortierellomycota were the main fungi in the three groups, accounting for more than 90% of the total abundance (Fig. 5A). Compared with the NC group, the abundance of Ascomycota in the Chili-S. rugosoannulata group after cultivating S. rugosoannulata significantly decreased, while the abundance of Basidiomycota, and Mortierellomycota increased significantly (p < 0.05). Compared with the NC group, the abundance of Chytridiomycota in the Chili-S. rugosoannulata group after cultivating S. rugosoannulata significantly decreased (p < 0.05). Compared with the NC group, the abundance of Ascomycota in the Rice-S. rugosoannulata group after cultivating S. rugosoannulata significantly decreased, while the abundance of Chytridiomycota and Basidiomycota increased significantly (p < 0.05) (Figs. 5B–5E).

Figure 5 Rhizosphere fungal community structure at phylum level after cultivating S. rugosoannulata in different croplands.

(A) Relative abundance of rhizosphere fungal at the genus level after cultivating S. rugosoannulata; (B) percentage of Ascomycota in each sample from the three groups after cultivating S. rugosoannulata; (C) percentage of Chytridiomycota in each sample from the three groups after cultivating S. rugosoannulata; (D) percentage of Basidiomycota in each sample from the three groups after cultivating S. rugosoannulata; (E) percentage of Mortierellomycota each sample from the three groups after cultivating S. rugosoannulata; (F) Relative abundance of rhizosphere fungal at the genus level after cultivating S. rugosoannulata; (G) percentage of Stropharia in each sample from the three groups after cultivating S. rugosoannulata; (H) percentage of Myrmecridium in each sample from the three groups after cultivating S. rugosoannulata; (I) percentage of Chaetosphaeria in each sample from the three groups after cultivating S. rugosoannulata; (J) percentage of Scytalidium each sample from the three groups after cultivating S. rugosoannulata. Data are mean ± SD (n = 6) and analyzed by one-way ANOVA. *p < 0.05.

At the genus level, the fungal community composition in the rhizosphere after cultivating S. rugosoannulata is shown in Fig. 5F. After cultivating S. rugosoannulata, compared with the NC group, the abundance of Stropharia and Myrmecridium in the Chili-S. rugosoannulata group was significantly increased, while the abundance of Chaetosphaeria and Scytalidium was significantly reduced (p < 0.05). Rice-S. rugosoannulata group significantly increased the abundance of Stropharia, but significantly decreased the abundance of Chaetosphaeria and Scytalidium (p < 0.05) (Figs. 5G–5J).

Function prediction of bacteria and fungi produced in different croplands

We found that soil organic carbon and available potassium had the greatest impact on the entire rotation process of cultivating S. rugosoannulata (Fig. 6A). We conducted a Pearson correlation analysis on the soil rhizosphere microorganisms at the phylum level and the amino acid content in S. rugosoannulata. We found that Chloroflexi, Nitrospirae, Acidobacteria and Verrucomicrobia are positively correlated with most amino acid content, while Bacteroidetes, Firmicutes, Actinobacteria and Proteobacteria are negatively correlated with most amino acid content (Fig. 6C). We used soil properties as environmental factors, and performed RDA analysis with soil rhizosphere fungi, and found that available phosphorus and available potassium had the greatest impact on the entire rotation process of cultivating S. rugosoannulata (Fig. 6B). We conducted a Pearson correlation analysis on the soil rhizosphere fungi at the phylum level and the amino acid content in S. rugosoannulata. We found that Rozellomycota, Chytridiomycota and Mortierellomycota are positively correlated with most amino acid content, while Ascomycota are negatively correlated with most amino acid content (Fig. 6D).

Figure 6 RDA analysis and correlation analysis of bacteria and fungi at the phylum level.

(A) RDA analysis between soil rhizosphere bacteria and environmental factors. (B) RDA analysis between soil rhizosphere fungi and environmental factors. (C) Correlation analysis of soil rhizosphere bacteria at phylum level and S. rugosoannulata amino acid content S. rugosoannulata. (da) Correlation analysis of soil rhizosphere fungi at phylum level and S. rugosoannulata amino acid content S. rugosoannulata.

Finally, we found that compared with the NC group, the microbes in the Chili-S. rugosoannulata group (Fig. 7A) and the Rice-S. rugosoannulata group (Fig. 7B) were two-component system, Microbial metabolism in diverse environments, Quorum Sensing and ABC transporters related functional genes are significantly enriched. We conducted a predictive analysis of the FUNGuild between different groups and found that compared with the NC group, the fungi in the Chili-S. rugosoannulata group (Fig. 7C) and the Rice-S. rugosoannulata group (Fig. 7D) were Plant Parasite, Dung Saprotroph and Soil Saprotroph related functional genes are significantly enriched.

Figure 7 Prediction of microbial Tax4Fun functional genes in different croplands.

(A) Differential bacterial functional genes between NC group and Chili-S. rugosoannulata group. (B) Differential bacterial functional genes between NC group and Rice-S. rugosoannulata group. (C) Differential fungi functional genes between NC group and Chili-S. rugosoannulata group. (D) Differential fungi functional genes between NC group and Rice-S. rugosoannulata group.

Discussion

This experiment explored the effect of idle cropland of two crops, chili and rice, on the cultivation of S. rugosoannulata. The results showed that this cultivation method significantly increased the yield and amino acid content of S. rugosoannulata and significantly changed the soil microbial community structure.Generally speaking, soil properties have a great influence on mushroom cultivation (Liu et al., 2021a). Our experiments have proved that the two crop rotation ways have significantly increased the total nitrogen content of the soil and promoted the growth of S. rugosoannulata. From the results of our experiments, it was found that total nitrogen in the soil before cultivation of S. rugosoannulata was significantly elevated after the harvest of both crops compared to the open field. Therefore, we think that the increase of total nitrogen in soil may be a critical factor in promoting the growth of S. rugosoannulata, which was also verified in the experiments of Esmaeilzadeh-Salestani et al. (2022), nitrogen in the soil is taken up and transformed by ammonium transporters (AMTs) in plants and ultimately promote plant growth. This is consistent with prior research findings. Due to the rise in microbial variety in rice-planted fields, the ability of microbial nitrogen fixation has improved, resulting in an increase in the soil’s nitrogen content (Chakraborty & Islam, 2018). Also shown the same ability to manage soil microbes was pepper as a crop rotation. After planting pepper, the relative number of bacteria and fungi in the soil increased dramatically, resulting in an increase in the soil’s nitrogen concentration (Chen et al., 2021). Nitrogen in the soil can be taken and transformed by plant roots into amino acids, structural proteins, and genetic material, according to studies (Kerru et al., 2020; Bloom et al., 2012). This may explain why the amino acid content of S. rugosoannulata in our experiment increased dramatically under two rotation patterns. We also noticed that in addition to total nitrogen, the phosphorus content in soil was also significantly increased due to the two rotation modes. The rice ecosystem had a huge impact on the composition of soil microbial community structure, and the change of microbial community also changed the phosphorus cycle in soil (Xu et al., 2019). Increasing the soil’s phosphorus concentration can considerably boost spore production and the colonization rate of arbuscular mycorrhizal fungi (Lin et al., 2020). In our experimental results, both phosphorus levels were significantly higher in the soils of agricultural fields planted with chili and rice. It is due to this mechanism that a high nitrogen content in the soil can promote the growth of S. rugosoannulata and improve their quality and yield. Other than that, this may be due to the fact that different farming patterns regulates the structure of soil microbial community and promotes the conversion of phosphorus to available phosphorus. We discovered a significant increase in soil nutrient content in paddy fields, which may be a result of significant changes in microbial communities within the paddy fields’ anaerobic environment  (Breidenbach et al., 2016). In addition, the genome sequencing of S. rugosoannulata revealed that it possesses powerful bioremediation and lignin degradation capabilities, so the secondary metabolites that can be produced can significantly improve soil nutrient levels (Yang et al., 2022). It has been demonstrated that the used mushroom stock remaining after the fruiting bodies have been harvested adds many nutrients to the soil  (Gong et al., 2018).

There are many ways in which soil rhizosphere microorganisms play a role in promoting crop growth (Maddalwar et al., 2021), for example, fix nutrients in the soil, improve crop utilization, metabolize plant auxins and avoid pathogen infection (Vejan et al., 2016). According to our and previous research, the composition and structure of soil microorganisms are associated with soil nutrient levels (Mishra, Singh & Singh, 2022). Esmaeilzadeh-Salestani et al. (2022) experiment also proved our conjecture. Their research revealed that increasing soil organic carbon content was advantageous for increasing microbial diversity, whereas rotation was more favorable to preserving soil organic matter and maintaining microbial activity (Esmaeilzadeh-Salestani et al., 2021). Therefore, we investigated the effects of S. rugosoannulata on soil microbial structure under two rotation modes. Studies have found that Chloroflexi has the genomic potential of using formate, which can rapidly carry out the carbon cycle and improve soil organic matter and nutrients (McGonigle, Lang & Brazelton, 2020). Therefore, we speculate that in our experiment, it may be due to the increase in Chloroflexi abundance, thereby improving soil fertility and ultimately promoting the growth of S. rugosoannulata. Sphingomonas has outstanding performance in remediate soil pollution and promoting plant growth. Recent studies have found that Sphingomonas can produce plant growth hormones such as gibberellin and indole acetic acid, which can promote plant growth (Asaf et al., 2020). Our experiment also found that the abundance of Sphingomonas in agricultural soils after planting chili and rice significantly increased, which also verified that Sphingomonas can promote crop growth. Gemmatimonas is a phototrophic bacteria based on chlorophyll (Zeng et al., 2020). Light can provide more energy for its growth and metabolism, thereby increasing the content of soil organic matter (Koblížek et al., 2020). Similarly, Gemmatimonas was significantly increased in the Rice-S. rugosoannulata group, which promoted the growth of S. rugosoannulata. In our experiment, Acidobacteria increased significantly in the Rice-S. rugosoannulat group. Genomic studies have shown that Acidobacteria has the ability to convert inorganic nitrogen into organic compounds. In addition, Acidobacteria can secrete extracellular polysaccharides, which can change the properties of rhizosphere soil and promote crop nutrient absorption. It may be that Acidobacteria regulates soil properties and ultimately promotes the growth of S. rugosoannulat. By fungal sequencing analysis we found that the expression abundance of Mortierella in Chili-S. rugosoannulata and Rice-S. rugosoannulataI group significantly increase. Studies have shown that Mortierella can be used as a biological control agent to reduce the occurrence of root rot (Pimentel et al., 2020). In addition, Cladosporium, which is significantly increased in Chili-S. rugosoannulata and Rice-S. rugosoannulataI group, can metabolize and release two volatile organic compounds (2-methyl-butyraldehyde and 3-methyl-butyraldehyde), which can significantly enhance crop growth and roots development (Liu et al., 2021b). In our experiment, it also promoted the growth of S. rugosoannulata. After S. rugosoannulata was cultivated, the significant increase in Stropharia abundance in Chili-S. rugosoannulata and Rice-S. rugosoannulataI group should be due to the fact that the yield of S. rugosoannulata in the soil was significantly higher than that of the NC group. Mycotoxins is a fungus that can produce toxins (Rajachan et al., 2017).

Through gene function analysis of bacteria, we found that the two-component system, Microbial metabolism in diverse environments, was significantly enriched in Chili-S. rugosoannulata and Rice-S. rugosoannulata groups. Bahn et al. (2006) find the two-component system can cascades control fundamental cellular functions related to fungal growth. Most two-component system-related signaling proteins are essential for fungal growth (Virginia et al., 2000). Microbial metabolism-related genes regulate bacterial metabolism and promote bacterial production of growth-promoting factors to regulate crop growth (Larsbrink et al., 2017). This also indicates that the soil microbial community structure was greatly regulated after cultivation of S. rugosoannulata, and that the microbial growth-promoting function was improved. Thus, we believe that mushroom cultivation in winter on vacant cropland can not only improve its quality and yield, but also increase soil microbial diversity, improve soil fertility, and lay a good foundation for a new round of crops in the coming year.

Conclusions

In this study, we systematically investigated changes in soil microbial communities in three cropping patterns: blank cropland, Chili-S. rugosoannulata and Rice-S. rugosoannulata. The results showed that the two crop rotation modes, Chili-S. rugosoannulata and Rice-S. rugosoannulata, significantly increased the yield, quality, and amino acid content of S. rugosoannulata, improved soil quality. Notably, S. rugosoannulata cultivation significantly increased the abundance of beneficial microorganisms such as Chloroflexi, Cladosporium and Mortierella and reduce the abundance of Botryotrichumin and Archaeorhizomyces. This leads us to believe that S. rugosoannulata cultivation in cropland can improve soil properties, regulate the community structure of soil microorganisms, increase the expression abundance of beneficial organisms and ultimately improve the S. rugosoannulata yield and lay a good foundation for a new round of crops after this edible mushroom cultivation.

Supplemental Information

Data S1 Raw data on yield, soil properties and amino acid content (Figures 1, 2 and Table 1)

Click here for additional data file.

We thank the reviewers and editors for their constructive comments, which significantly improved our manuscript.

Additional Information and Declarations

Competing Interests

Author Contributions

Data Availability

The authors declare there are no competing interests.

Shaojun Tang conceived and designed the experiments, prepared figures and/or tables, and approved the final draft.

Tingting Fan performed the experiments, prepared figures and/or tables, and approved the final draft.

Lei Jin performed the experiments, authored or reviewed drafts of the article, and approved the final draft.

Pin Lei performed the experiments, authored or reviewed drafts of the article, and approved the final draft.

Chenxia Shao analyzed the data, authored or reviewed drafts of the article, and approved the final draft.

Shenlian Wu analyzed the data, authored or reviewed drafts of the article, and approved the final draft.

Yi Yang analyzed the data, authored or reviewed drafts of the article, and approved the final draft.

Yuelin He performed the experiments, authored or reviewed drafts of the article, and approved the final draft.

Rui Ren performed the experiments, authored or reviewed drafts of the article, and approved the final draft.

Jun Xu conceived and designed the experiments, authored or reviewed drafts of the article, and approved the final draft.

The following information was supplied regarding data availability:

The raw sequences are available at NCBI Sequence Read Archive: PRJNA787767.

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
