# Peer review of "Soil microbial diversity and functional capacity associated with the production of edible mushroom Stropharia rugosoannulata in croplands"

_PeerJ, doi:10.7717/peerj.14130_

## Round 0.1 · original submission · Major Revisions

I agree with the reviewers that the article can be considered for acceptance after major revisions.

·

Basic reporting

no comment

Experimental design

no comment

Validity of the findings

no comment

Additional comments

The authors explored the effects of Stropharia rugosoannulata in rotation with chili and rice on soil properties and soil microbial community structure. The yield and quality traits of S. rugosoannulata among experimental groups were also evaluated and compared. This work is of great interest for S. rugosoannulata cultivation and soil amelioration. The materials and methods used are appropriate for the scope of the study. Results have been presented and discussed properly.

Minor Points:
Line 86: the size of three experimental plots needs to be detailed.
Line 174-200: it is suggested to add several references for data analysis methods and software.
Line 202-203: the number of fruiting bodies chosen for phenotypic evaluation should be mentioned.
Maybe due to version issues, the background of Fig3 and Fig is dark and unclear.
In Discussion section, some overstatements need to be tuned down, such as line 350-351, Line 364.

I have noticed a few grammatical and editorial errors that need to be improved. Please proofread the full manuscript.
Line 16: delete “(S. rugosoannulata)”; Line 32&34: “Stropharia rugosoannulata” in should be “S. rugosoannulata”; Line 108, 202, 239…. “S. rugosoannulata” should be in italic. Please unified the whole text.
Line 24: “Chili-S. rugosoannulata” -> “Chili-S. rugosoannulata”.
Line 28: “we” -> “We”.
Line 50: “mycelim” -> “mycelium”.
Line 97: “3 percent” -> “3%”.
Line 101: the sentence “The growth of the fungus to fruiting bodies took three-four months.” needs to be reworded.
Line 106: “are” -> “were”.
Line 121: “12.0 ~ 12.5 min” -> “12.0-12.5 min”
Line 172: space missing between value and units.
Line 207: please revise the p value in uniform format (p, lower case and italic) in the whole text.
Line 211-215: what is the “mushroom legs” and “legs of S. rugosoannulata”? Please revise it to the academic terms.
Line 219: the unit of S. rugosoannulata yield is suggested to be kg/m2 or kg/ha.
Line 235: “Soil” -> “soil”.
Line 245: “Figure 3A” -> “Figure 4a”.
Line 261: “microorganisms” is suggested to be “fungi”.
Line 292-295: this sentence is confused and needed to be reworded.
Line 394: “I Notably, S. rugosoannulata” -> “Notably, S. rugosoannulata”.

Reviewer 2 ·

Basic reporting

The structure of the article is relatively complete, the data is detailed and reliable, and the provided pictures and table data are relatively standard, but the language of the article still needs to be strengthened

Experimental design

The experimental design is complete, which supports the conclusion of the paper

Validity of the findings

The authors provide insight into the impact of field cultivation of Stropharia mushrooms on soil quality and soil microbial communities. This is useful and I feel more studies along this topic are required to encourage an increase in practices that can not only improve soils, but also increase the agricultural outputs for a unit of land.

Additional comments

1.Lines32, The Latin genus name of the same species should be abbreviated on the second or subsequent occurrences.Please check the full text.
2.Line 101,“it seems to be a good choice to cultivate S. rugosoannulata by planting these two crops in winter on unused acreage”.Do you mean planting these two crops in winter? Please check it.
3.Line 101, The fungus here should be plural(fungi).
4.Line 104, What is the five-point sampling method? It is best to describe in detail or provide references.
5.Lines 115-116.“Using High-performance liquid chromatograph (Model 1100, 116 Agilent, USA) analysis and detection of amino acids in biological tissues”is confusing to read.
6.Line 118, “NaH2PO4”,numbers should be subscripted.
7.Line 120, (B: 57%) ~100%)?
8.Line 202,S. rugosoannulata,should be in italics.
9.Line 235,“Soil” should be “soil”
10.Line 239,S. rugosoannulata,should be in italics.
11.Lines 244,263, 347,...,Species at phylum level do not require italics.Please check the full text
12.Lines 256,258, Candidatus_Solibacter,should be written correctly,and species should be abbreviated on the second.
13.Line 334, S. rugosoannulata should be in italics.

Reviewer 3 ·

Basic reporting

This is an interesting and original research related to the enhancement originated by the cultivation of the edible mushroom Stropharia rugosoannulata in terms of soil fertility in agroecosystems where two crops have been cultivated compared with non-cultivated soils. Additionally, information related to the microbial soil communities originated by the cultivation of this mushroom and their potential functions are described in detail. There has been previous works related with the topic, however these works have studied the changes originated by the cultivation of this important mushroom in China, in other conditions, basically cultivation under the forest shade. The rotative system crop-Stropharia rugosoannulata has received little attention so far worldwide. Therefore, I think that the manuscript have the possibility to be published in the prestigious Journal PeerJ after minor revisions. These include:

1. Lines32,34,39..., “Stropharia rugosoannulata” should be “S. rugosoannulata”, and consistent elsewhere in the full text.
2. Line 35, “increase the expression abundance of beneficial organisms and ultimately improve the Stropharia rugosoannulata yield and lay a good foundation for a new round of crops in the coming year.” should be “increase the expression abundance of beneficial organisms and ultimately improve the Stropharia rugosoannulata yield and lay a good foundation for a new round of crops after this edible mushroom cultivation.”
3. Line 59, “mushroom culture” should be “mushroom cultivation”.
4. Line 90, A mushroom is not "planted" because is not a plant, but cultivated!
5. L174-L186, The details of sequences data for the sample should be added such as the sequences number of each sample.
6. L188, What’s the RNA analysis? Delete the CCA.
7. Line 214, “(Fig. 1B, C)” should be “(Fig. 1b, c)”. The serial number of the figure should be consistent with the annotation in the figure.
8. Line 216, was, please use past tense throughout the text.
9. Line 225, “amino acids increase substantially” should be “amino acids increased substantially”. please use past tense throughout the text.
10. Line230, “organic carbon” should be “soil organic carbon”. The same description in all places also need to be corrected
11. Line 341, please describe with more detail, not only with an "etc." this is a crucial part of your work!
12. Line 352, “repairing” should be “remediate”
13. Line 367, “Mortierella” should in italics.

Experimental design

The experimental design is basically reasonable and the details of method is present clearly.

Validity of the findings

The robust data supports the conclusion well.

Additional comments

Nothing

---

## Round 0.2 · accepted · Accept

The authors have made revisions in accordance with their comments, and the current manuscript version is recommended for publication in PeerJ.

·

Basic reporting

no comment

Experimental design

no comment

Validity of the findings

no comment

Additional comments

The author answered all my concerns.

Reviewer 2 ·

Basic reporting

Clear and unambiguous, professional English used throughout.

Experimental design

Research question well defined, relevant & meaningful. It is stated how research fills an identified knowledge gap.

Validity of the findings

All underlying data have been provided; they are robust, statistically sound, & controlled.

Reviewer 3 ·

Basic reporting

In my opinion, this manuscript has been revised properly and is worthy of being published.

Experimental design

The experimental design is OK.

Validity of the findings

The data support the conclusion.

Additional comments

Nothing